# Effects of Ultraviolet Radiation on Sediment Burial Parameters and Photo-Oxidative Response of the Intertidal Anemone *Anthopleura hermaphroditica*

**DOI:** 10.3390/antiox11091725

**Published:** 2022-08-31

**Authors:** Víctor M. Cubillos, Javier A. Álvarez, Eduardo Ramírez, Edgardo Cruces, Oscar R. Chaparro, Jaime Montory, Carlos A. Spano

**Affiliations:** 1Instituto de Ciencias Marinas y Limnológicas, Facultad de Ciencias, Universidad Austral de Chile, Valdivia 5090000, Chile; 2Laboratorio Costero de Recursos Acuáticos de Calfuco, Facultad de Ciencias, Universidad Austral de Chile, Valdivia 5110566, Chile; 3Centro de Investigaciones Costeras, Universidad de Atacama (CIC-UDA), Avenida Copayapu 485, Copiapó 1530000, Chile; 4Centro i~mar, Universidad de Los Lagos, Casilla 557, Puerto Montt 5480000, Chile; 5Departamento de Oceanografía Biológica, Ecotecnos S.A., Limache 3405, Viña del Mar 2520000, Chile

**Keywords:** antioxidant metabolism, behavior, oxidative damage, sea anemone, sediment burrowing, UVB radiation

## Abstract

*Anthopleura hermaphroditica* is an intertidal anemone that lives semi-buried in soft sediments of estuaries and releases its brooded embryos directly to the benthos, being exposed to potentially detrimental ultraviolet radiation (UVR) levels. In this study, we investigated how experimental radiation (PAR: photosynthetically active radiation; UVA: ultraviolet A radiation; and UVB: ultraviolet B radiation) influences burrowing (time, depth and speed) in adults and juveniles when they were exposed to PAR (P, 400–700 nm), PAR + UVA (PA, 315–700 nm) and PAR + UVA + UVB (PAB, 280–700 nm) experimental treatments. The role of sediment as a physical shield was also assessed by exposing anemones to these radiation treatments with and without sediment, after which lipid peroxidation, protein carbonyls and total antioxidant capacity were quantified. Our results indicate that PAB can induce a faster burial response compared to those anemones exposed only to P. PAB increased oxidative damage, especially in juveniles where oxidative damage levels were several times higher than in adults. Sediment offers protection to adults against P, PA and PAB, as significant differences in their total antioxidant capacity were observed compared to those anemones without sediment. Conversely, the presence or absence of sediment did not influence total antioxidant capacity in juveniles, which may reflect that those anemones have sufficient antioxidant defenses to minimize photooxidative damage due to their reduced tolerance to experimental radiation. Burrowing behavior is a key survival skill for juveniles after they have been released after brooding.

## 1. Introduction

Many infaunal organisms depend upon burrowing for survival, increasing their levels of food capture [1,2,3], generating physical protection against predators or minimizing physiological stress, especially in highly unstable environments [4,5,6]. Estuaries are considered highly stressful areas because tidal cycles generate pronounced environmental changes and high levels of physiological stress in individuals living there [7,8,9], for example, by exposing sessile or sedentary organisms to the detrimental action of ultraviolet-B radiation (UVB; 280–320 nm), especially when low tide coincides with the daily peak in environmental radiation [10]. The absorption of UVB by different chromophores (i.e., DNA, amino acids, melanin and metabolites) indirectly exacerbates the generation of reactive oxygen species (ROS) [11,12,13,14] that are normally formed as byproducts of oxygen reduction in the cell. Consequently, ROS over-accumulation can induce structural and functional changes in lipids, proteins and DNA [15]. ROS attack polyunsaturated fatty acids; they are incorporated into lipids to form lipid peroxides [16], which can induce the generation of very reactive long-lived byproducts such as aldehydes (malondialdehydes, MDA; 4-hydroxynonenal, HNE and keto fatty acids) [17,18,19]. Consequently, their generation decreases membrane permeability, inducing electron leakiness in the cell [19]. Proteins with specific amino acids (e.g., proline, arginine, lysine and threonine) can be rapidly affected by ROS, inducing the formation of carbonyl groups [20]. Secondary reactions between specific amino acid side chains and byproducts of lipid peroxidation can induce the formation of carbonyl groups in proteins [20]. Subsequently, protein carbonylation reduces the catalytic activity of enzymes and induces further breakdown of proteins [21]. This results in a crosslinked reaction product, increasing the susceptibility of amino acids and proteins to degradation [18]. Consequently, oxidative damage can trigger physiological [13], anatomical [22], reproductive [23] and ecological [24,25] problems in marine invertebrates, which can cause death in severe cases [26]. However, aerobic organisms can synthesize enzymatic (e.g., superoxide dismutase (SOD), catalase (CAT), glutathione peroxidase (GPx), glutathione reductase (Gr)) and non-enzymatic (glutathione (GSH), oxidized glutathione (GSSG)) antioxidants that minimize cellular damage. Antioxidants can transfer electrons to these oxidizing agents (ROS) by neutralizing them and reducing chain reactions that can cause cell damage [27], reaching cellular homeostasis.

In southern Chile, the sea anemone *Anthopleura hermaphroditica* inhabits estuaries with high levels of ultraviolet B radiation (UVB, 280–315 nm) radiation during summer, which can exceed 50% of values recorded in areas of the Northern Hemisphere for the same latitude and season of the year [28]. This temperate anemone burrows in the sediment, spreading only its tentacles over the surface [29], allowing it to expose its symbiotic dinoflagellate (*Phylozoon anthopleurum*) [30] to environmental radiation. This species incubates its young inside the gastrovascular cavity (GVC) and releases them into the benthos once they reach the tentaculate juvenile stage [31]. Thus, juveniles may be exposed to significantly higher levels of UVB radiation than those experienced inside the incubation cavity, which can generate elevated oxidative damage to them, considering that the initial stages of development are highly affected by UVB radiation [22,32].

In natural conditions, this anemone is found buried in the sediment, projecting only the tentacular crown on the substrate; however, there is no evidence about the reason for this behavior. Although UVB radiation generates cytotoxic effects, we hypothesized that the sediment could provide physical protection against UVB radiation after burial in the sediment, minimizing photo-oxidative damage and the antioxidant response. We also postulated that juvenile anemones are much more susceptible to photo-oxidative damage than adults when exposed to different levels of experimental radiation. This is supported by the fact that under natural conditions, juveniles develop inside the GVC of adults buried in the sediment, always exposed to low radiation conditions. Thus, leaving the GVC and facing an environment with high levels of radiation could generate higher levels of oxidative damage and antioxidant response than in adult individuals. Considering burial in the sediment as a natural process in the life cycle of this cnidarian, it is likely that adult and juvenile anemones exposed to UVB bury themselves in the sediment at higher rates than those exposed without the presence of UVB, as an evasive response to cell damage.

Considering that anemones are conspicuous organisms that play an important role in different aquatic ecosystems supporting different organisms through symbiotic relationships [33,34,35] and being part of different trophic webs [36,37,38], it is important to understand the role of sediment in the photooxidative response of *A. hermaphroditica* as an important component of the Quempillén estuary.

In order to test the role of sediment in the photooxidative response of anemones and the elevated susceptibility of juveniles to experimental radiation, adults and juveniles of *A. hermaphroditica* were exposed to a combination of UVB, UVA and PAR radiation with and without sediment and oxidative damage (lipids and proteins), and total antioxidant capacity was quantified. In order to test the effects of experimental radiation on the behavioral response of anemones, juveniles and adults were placed over the substrate under a combination of UVB, UVA and PAR; time, depth and speed were assessed using image analysis.

## 2. Materials and Methods

### 2.1. Animal Collection and Maintenance

Approximately 200 individuals of *A. hermaphroditica* and their sediments (60% gravel and 40% sand) were obtained from the Quempillén estuary (41°52′17.36″ S, 73°46′1.02″ W, Figure 1) during the austral summer season, placed in plastic boxes (80 × 45 × 30 cm) and then transferred to the Laboratorio Costero de Recursos Acuáticos de Calfuco (39°46′50″ S, 73°23′34″ W) at the Universidad Austral de Chile (Valdivia, Chile).

Anemones were acclimated in an aquarium with sediment (collected from the sampling site) where circulating seawater (35 ppt, 13 °C) was provided, with constant aeration for an acclimation period of 3 days and a photoperiod of 12:12 h with photosynthetically available radiation (PAR; 400–700 nm) at an irradiance of 1000 µmol m^−2^ s^−^^1^. Prior to the experiments, anemones were separated into two size ranges as follows: juveniles (1–2 cycles of tentacles; <2 mm pedal disc diameter) and adults (>3 cycles of tentacle; >4 mm pedal disc diameter).

### 2.2. Experimental Treatments

In order to determine the effect of radiation on the substrate burial response, adults and juveniles of *A. hermaphroditica* were exposed to the following experimental radiation conditions: PAR (400–700 nm), PAR + UVA (315–700 nm) and PAR + UVA + UVB (280–700 nm). Three rectangular transparent glass aquaria 30 cm long, 10 cm high and 5 cm wide were set up for the experiment, with a seawater circulation system activated by a submersible pump (220 L h^−1^) (Figure 2).

The sediment collected in the field was deposited in each aquarium to a thickness of 4 cm. Anemones were deposited individually on the sediment of the aquaria, which were immediately covered with Clear 226-UV (Chris James Lighting Filters, London, UK), Folanorm SF-AS (0.13 mm; Lüerssen Grafische Vertriebs GmbH, Germany) and cellulose acetate filters (CDA; 0.13 mm; Grafix^®^, Maple Heights, OH, USA) to generate the radiation treatments P (400–700 nm), PA (315–700 nm) and PAB (280–700 nm), respectively. PAR fluorescent tubes (Philips Day Light, Philips, Eindhoven, The Netherlands), UVA (Philips TL40 W, Philips, Eindhoven, The Netherlands) and UVB (Phillips TL20, Philips, Eindhoven, The Netherlands) were placed 30 cm above the experimental aquaria to simulate the natural irradiance conditions registered in the environment [10].

Anemones were thus exposed to an irradiance of 1000 µmol m^−2^ s^−1^, 20 W m^−2^ and 2.3 W m^−2^ of PAR, UVA and UVB, respectively, for 3 h. The irradiance of the lamps was quantified using a portable radiometer (PMA-2100, Solar Light, Glenside, PA, USA) fitted with a PAR sensor (PMA-2132 model), UVA sensor (PMA-2110) and UVB sensor (PMA-2106). The experiment was replicated 11 times for each developmental stage. New individuals were used for each experiment.

### 2.3. Behavioral Responses

In order to quantify sediment burial responses (time, depth and speed) under different experimental radiation treatments, adults and juveniles of *A. hermaphroditica* were recorded using a set of four digital cameras connected to a digital video recorder. The maximum recording time was three hours. The burial parameters were defined as follows:

#### 2.3.1. Burial Time

Burial time was defined as the time elapsed from when the adult and juvenile anemones were placed on the surface of the sediment until the individuals were buried in the sediment in response to the respective radiation treatments.

#### 2.3.2. Burial Depth

Burial depth was measured as the distance between the sediment surface and the distal end of the pedal disc of polyps. Analysis of the images recorded during the burial process allowed calculating the depth to which the body column of adults and juveniles of *A. hermaphroditica* were buried in the sediment. Burial depth was measured using as reference a graduated ruler (mm) previously attached to each experimental aquarium.

#### 2.3.3. Burial Speed

Burial speed was determined in adults and juveniles of *A. hermaphroditica* exposed to different radiation treatments (P, PA and PAB) from the ratio of depth to burial time. The speed data were expressed in mm h^−1^.

### 2.4. Cellular Responses

The oxidative damage and the antioxidant capacity of adults and juveniles of *A. hermaphroditica* were quantified using the same aquaria and experimental radiation conditions indicated previously with and without sediment (Figure 2(B.1,B.2)). Anemones were deposited individually in each aquarium and then exposed to P, PA and PAB radiation treatments for 3 h. The experiment was replicated four times for adults and five times for juveniles. After the three experimental hours, individuals from both conditions (with and without sediment) were immediately placed in liquid nitrogen. The tissue was ground using a pestle and mortar to which liquid nitrogen was added. The fine powder obtained during grinding was kept in Eppendorf tubes at −80 °C until oxidative damage (lipid peroxidation and protein carbonyl formation) and total antioxidant capacity were quantified.

#### 2.4.1. Oxidative Damage Assays

Levels of lipid oxidation were determined by malondialdehyde (MDA) quantification using the colorimetric method of thiobarbituric acid reactive substance (TBARS), according to the methodology described by Salama, A. and Pearce, R. [39]. Thirty g (fresh weight, FW) of ground tissue was homogenized with 350 μL of trichloroacetic acid (TCA, 0.1%, Merck KGaA, Darmstadt, Germany) in an Eppendorf tube and centrifuged for 10 min at 13,000 RPM (4 °C) (Labnet, Prism R, Edison, NJ, USA). A volume of 200 µL was extracted from the supernatant and homogenized with 500 µL of TCA (20%) + TBA (thiobarbituric acid, 0.5%) solution in a new Eppendorf tube and then incubated in a ThermoMixer (Eppendorf, Hamburg, Reinbek, Germany) at 80 °C for 30 min. Next, the tubes were incubated on ice for five minutes and centrifuged again for five minutes at 13,000 RPM (4 °C). Finally, malondialdehyde (MDA) levels were quantified by measuring the absorbance at 532 nm of 200 µL of the supernatant in a plate reader (Anthos, Biochrom Ltd., Cambridge, UK). The concentration of MDA was determined using the molar extinction coefficient, and the results were expressed in nm MDA g^−1^ FW.

Protein carbonyl levels were estimated by the method of dinitrophenylhydrazine (DNPH) used by Levine [40]. An amount of 80 mg of previously ground tissue was homogenized with 2 mL of a 50 mM phosphate buffer solution (0.1 mM EDTA and 1% PVPP polyvinyl polypyroline) at pH 7.0 and centrifuged for 20 min at 13,500 rounds per minute (RPM) (4 °C). Subsequently, 200 μL of the supernatant was homogenized with 200 μL 20% TCA, incubated for 30 min (−20 °C) and centrifuged at 13,500 RPM (4 °C). The resulting supernatant was removed to reserve the pellet. Then 300 μL DNPH solution (100 mM in 2 N HCl) was added to the pellet and incubated in the dark for 1 h. Next, the mixture was precipitated with 500 μL TCA (20%) and incubated again for 15 min at −20 °C. The resulting mixture was centrifuged for 10 min at 13,200 RPM; then, the supernatant was removed carefully to reserve the pellet. The resulting pellet was re-suspended with 500 µL ethanol–ethyl-acetate (1:1) solution and centrifuged for 10 min at 13,200 RPM (4 °C), removing the supernatant again. Next, 2 mL guanidine HCL (6 M) in 20 mM sodium phosphate (pH 6.8) was added to each sample and then centrifuged for 30 min at 9000 RPM (4 °C). Finally, protein carbonyl levels were determined at 380 nm using a Zenyth 100 plate reader (Anthos, Biochrom Ltd., Cambridge, UK), and their concentrations were determined using the molar extinction coefficient.

#### 2.4.2. Total Antioxidant Capacity

The method described by Fukumoto, L. and Mazza, G. [41] was applied for the determination of total antioxidant capacity, which uses 2,2 diphenyl-1-picrylhydrazyl (DPPH). For this quantification, 60 mg of tissue was homogenized in 1 mL of a 70% acetone solution, then the tubes with their content were incubated in an ultrasonic bath for 2 h. Then they were centrifuged for 5 min at 9000 RPM at 4 °C and kept under a fume hood for 90 min, allowing the acetone to evaporate. Then, 200 μL DPPH was added to each tube. Finally, the absorbance was quantified every 10 min for 120 min using a Zenyth 100 plate reader (Anthos, Biochrom Ltd., Cambridge, UK) at 520 nm. That information generated a standard curve of 6-hydroxy-2,5,7,8-tetramethylchromane-2-carboxylic acid (Trolox) for calculating the total antioxidant capacity, expressed as mg Trolox Eq FW g^−1^ of the sample.

### 2.5. Statistical Analysis

The assumptions of normality and homogeneity of variances of the burial parameters (time, depth and speed), as well as the levels of oxidative damage and total antioxidant capacity for juveniles and adults, were confirmed by Levene’s test and the Shapiro–Wilk test, respectively. When necessary, data transformation (Log_10_) was used to fulfill the assumptions of normality and homogeneity of variances. By using a two-way ANOVA, it was determined whether the development status and/or radiation treatment (P, PA and PAB) influenced burial behavior (i.e., time, depth and speed). Additionally, a three-way ANOVA was used to determine if the levels of oxidative damage (protein carbonyl and lipid peroxidation) in *A. hermaphroditica* were influenced by the developmental state (juveniles and adults), experimental radiation (P, PA and PAB) and level of protection provided by the sediment (presence and absence). Finally, a Tukey *a posteriori* test was used to identify the significant differences. Statistical analyses were carried out with SigmaPlot (Sigma Plot for Windows version 11, Systat Software Inc., Chicago, IL, USA). The significance level was defined at *p* < 0.05.

## 3. Results

### 3.1. Behavioral Responses

#### 3.1.1. Burial Time

The two-way ANOVA indicated that radiation treatment and development stage, as well as the interaction of these factors, significantly influenced the burial time of anemones in the sediment (Figure 3A, Table 1). Significant differences in burial time were observed between the developmental stages; juvenile anemones had a shorter burial time than adults (Figure 3A, Table 1). Additionally, experimental radiation treatments that included UV radiation (PA and PAB) had a significant effect in reducing the burial time of adults but not of juveniles (Figure 3A, Table 1).

#### 3.1.2. Burial Depth

The two-way ANOVA indicates that both developmental stages and radiation treatments significantly influenced the depth to which *A. hermaphroditica* was buried in the sediment (Figure 3B, Table 1). Particularly, the PAB radiation treatment caused adult anemones to bury significantly deeper (23%) than those exposed only to P and PA radiation treatments (Figure 3B, Table 1). Although juvenile anemones burrow to shallower depths than adult anemones, there were no significant differences in the depths that they burrowed when they were experimentally exposed to the P, PA and PAB radiation bands (Figure 3B, Table 1).

#### 3.1.3. Burial Speed

Our results indicate that radiation treatment significantly influenced the burrowing speed of *A. hermaphroditica* in the sediment (Figure 3C, Table 1). When adults and juveniles of *A. hermaphroditica* were exposed to the PAB radiation treatment, they increased their mean burial rate by 132% and 36%, respectively, compared to those exposed only to the P-radiation treatment (Figure 3C, Table 1). Radiation treatment PA and PAB showed significant differences in the burial speed of adult sea anemones compared to P (Figure 3C, Table 1). PAB radiation treatment of juveniles induced a significant increase in velocity compared to PA and P. Juvenile sea anemones showed higher burial speed than adults only under P radiation treatment.

### 3.2. Cellular Responses

#### 3.2.1. Lipid Peroxidation

The three-way ANOVA indicated that type of radiation treatment (*F*_(2,29)_ = 25.97, *p* < 0.001, Table 2), stage of development (*F*_(1,29)_ = 158.21, *p* < 0.001, Table 2) and the sediment condition (*F*_(1,29)_ = 12.87, *p* = 0.001, Table 2) all influenced lipid peroxidation levels in the sea anemone *A. hermaphroditica*. Tukey’s *a posteriori* test showed that the presence of UVB through PAB radiation treatment exerted a significant role in the generation of oxidative lipid damage, as opposed to what happens to anemones exposed to P and PA. Juvenile anemones are more susceptible to lipid peroxidation than adult anemones. The presence of sediment in the radiation exposure treatments significantly reduced the level of lipid photo-oxidation in experimental anemones (Table 2).

Only the interaction between the developmental stage and experimental radiation treatment significantly influenced the generation of lipid oxidation (*F*_(2,29)_ = 11.12, *p* < 0.001, Table 2); PAB generated a significant increase in the level of oxidative lipid damage in adult anemones. P, PA and PAB radiation significantly influenced lipid oxidation in juvenile anemones (Figure 4A,B; Table 2). Juveniles exposed with sediment to P, PA and PAB radiation increased their lipid peroxidation levels by four, five and four times, respectively, compared to adult anemones exposed to the same conditions (Figure 4A,B; Table 2). Similarly, juveniles exposed without sediment to P, PA and PAB increased their lipid peroxidation levels by two, three and three times, respectively, compared to adult anemones exposed to the same conditions (Figure 4A,B; Table 2).

#### 3.2.2. Protein Carbonyl

The three-way ANOVA indicated that the generation of protein carbonyl levels in *A. hermaphroditica* in this study were significantly influenced by the radiation type (*F*_(2,32)_ = 11.93, *p* < 0.001, Table 2), developmental stage (*F*_(1,32)_ = 351.27, *p* < 0.001, Table 2) and sediment condition (*F*_(1,32)_ = 8.84, *p* = 0.006, Table 2). PAB generated a significantly higher level of protein carbonyl than those observed under the PA and P radiation treatments (Figure 4C,D; Table 2). Juvenile anemones were significantly more susceptible to protein oxidation than adult anemones, independent of the radiation treatment (Figure 4C,D; Table 2).

Particularly, the presence or absence of sediment had a significant effect on the level of protein oxidation in adults and juveniles of *A. hermaphroditica* exposed to experimental radiation. The presence of sediment in the experimental aquaria protected anemones against protein oxidation, especially when they were exposed to PAB radiation treatment. The absence of sediment in the experimental aquaria produced significant differences in the level of protein carbonyls in adults and juveniles. Juveniles exposed to experimental radiation without sediment significantly increased the level of oxidative damage to proteins more than adults.

The interaction of the developmental stage and radiation treatment generated a significant impact on protein oxidation levels (Figure 4C,D; Table 2). Significant differences in carbonyl levels were observed when juveniles and adults were exposed to either PA or PAB. Adults, but not juveniles, showed significant differences in protein carbonyl levels when anemones with and without sediment were exposed to P radiation. A significant difference in protein carbonyl level of juvenile anemones was observed between P and PAB radiation treatment. Juveniles in sediment directly exposed to P, PA and PAB generated carbonyl levels 28, 9 and 10 times greater, respectively, than adults exposed under the same conditions (Figure 4C,D; Table 2). When juveniles without sediment were directly exposed to P, PA and PAB, they generated carbonyl levels 6, 10 and 10 times higher, respectively, than adults exposed to the same experimental radiations (Figure 4C,D; Table 2).

#### 3.2.3. Total Antioxidant Capacity

Three-way ANOVA indicated that the experimental radiation type (*F*_(2,34)_ = 5.67, *p* = 0.007, Table 2), developmental stage (*F*_(1,34)_ = 4.22, *p* = 0.048, Table 2) and sediment condition (*F*_(1,34)_ = 19.19, *p* < 0.001, Table 2) significantly influenced total antioxidant capacity in *A. hermaphroditica* (Figure 4E,F). Juveniles exhibited greater total antioxidant capacity than adults. PAB radiation treatment significantly increased the total antioxidant capacity response in *A. hermaphroditica* compared to individuals exposed to P. Adults exposed to experimental PAB radiation treatment generated significant differences in total antioxidant capacity only when compared to P-exposed anemones. Juvenile sea anemones showed higher antioxidant capacity levels when they were exposed to PA and PAB compared to P-exposed anemones.

The presence or absence of sediment significantly influenced the total antioxidant response in anemones. A significant difference in total antioxidant capacity was observed between adults and juveniles exposed to experimental radiation with sediment. The interaction of developmental stage and sediment condition significantly influenced the total antioxidant capacity response in *A. hermaphroditica* (*F*_(1,34)_ = 27.89, *p* < 0.001, Table 2).

## 4. Discussion

### 4.1. Burrowing Behavior against Experimental Radiation

UVB radiation is an abiotic factor known to control the distribution and abundance of different marine organisms in a bathymetric profile due to its negative effects on cells, with physiological and behavioral consequences [42]. Evasive responses to UVR include habitat selection processes [43,44], vertical/horizontal migration in the water column [45], burial in the sediment [46] and body covering with physical structures that minimize damage induced by direct or indirect exposure [47,48]. UVR evasive responses described previously in sea anemones include covering the body column with either gravel or debris and contraction of its body column [49].

In the Quempillén estuary, the sea anemone *A. hermaphroditica* inhabits the sediment, semi-buried, protecting its complete body column from direct UVR radiation, projecting only its tentacles over the surface. Our results demonstrate for the first-time burial in the sediment of the intertidal anemone *A. hermaphroditica* as an evasive response to UVB radiation. Similar avoidance responses were found in the mantis shrimp *Haptosquilla trispinosa*, which possesses negative phototaxis when exposed to UVB and UVA radiation, sheltering in its burrow constructed in the sediment [46]. Exposure of *A. hermaphroditica* adults to PAR + UVA and PAR + UVA + UVB results in a reduction in burial time. Our results indicate that juvenile anemones have increased susceptibility to radiation; thus, they keep their burial times almost constant under different combinations of PAR, PAR + UVA and PAR + UVA + UVB experimental radiation. This indicates that there is a need for juveniles to bury as quickly as possible to reduce radiation exposure times and thus minimize photo-oxidative damage. Previous observations in the intertidal anemone *Actinia tenebrosa* indicate that exposure times are determinant in the levels of DNA damage through the generation of CPDs, which can increase by approximately 200% and 300% in anemones exposed to PAR + UVA and PAR + UVA + UVB compared to those exposed only to PAR (control) for a period of 96 h [50]. Thus, adults and juveniles of *A. hermaphroditica* should increase their burial speeds, minimizing exposure time, especially when exposed to UVB radiation. Rapid evasion to UVB exposure was identified in early developmental stages (zoea I and zoea II) of the Patagonian crustacean *Cyrtograpsus altimanus*, which increased evasion speed by 50% compared to those exposed only to PAR radiation [51].

The ability of anemones to bury themselves in sediment appears to be inversely related to the development of body outgrowths. *A. hermaphroditica* tends to adhere to gravel or pebbles, preventing overexposure to the environment (i.e., avoiding desiccation; [52]). This adherence process is much more common in the rocky intertidal zone; therefore, the possibility of burrowing seems to be an adaptive response to the environment where the sediments are too fine to be attached to the body column. In adults, burrowing has not only been shown to minimize photo-oxidation during incubation, but also it seems to be a more successful strategy than sticking stones to the body wall, at least when comparing population densities reached in hard and soft-bottom habitats [29,31].

### 4.2. Cellular Response in A. hermaphroditica

The levels of lipid peroxidation and protein carbonyls observed in this study are lower than those previously reported for *A. hermaphroditica* in the Quempillén river estuary [10]. Our study demonstrates that exposure to PAR + UVA and PAR + UVA + UVB radiation generated a significant increase in the levels of oxidative damage to lipids and proteins in *A. hermaphroditica*. Similar results were observed in the intertidal anemone *Actinia tenebrosa*, where exposure to P, PA and PAB conditions generated lipid peroxidation levels of 3.8, 6.0 and 10 nmol mg FW^−1^, respectively [50]. In the same species, the levels of protein carbonyls were approximately 1.8, 5.0 and 6.0 for anemones exposed to P, PA and PAB, respectively, after 96 h of exposure. It was well established that the presence of photosensitizing molecules generates greater absorption of UVB radiation in the cell, producing photo-oxidation of different macromolecules [53]. In this study, higher levels of photo-oxidative damage were found in juvenile anemones than in adult anemones exposed to P, PA and PAB treatments, confirming their high susceptibility to photo-oxidation. The above is consistent with previous studies, showing high susceptibility of the initial developmental stages of aquatic organisms to UVB, which includes pluteus larvae of urchins [54], zoea I larvae of crustaceans [55,56,57], fish larvae [58,59] and amphibian larvae [60]. Meta-analysis studies support that early stages of development are much more prone to damage than adult organisms [61,62,63], mainly due to their limited energy reserve [64].

Prolonged exposure of larvae of the Antarctic urchin *Sterechinus neumayerii* to PAB radiation generates increases in the levels of lipid peroxidation and protein carbonyls by approximately 3-fold compared to larvae exposed to P radiation [65]. However, when larvae of this echinoderm were cultured under the Antarctic ice cover, the levels of abnormalities in PAB were like those generated under P radiation. Thus, the Antarctic ice cover may act as a physical barrier, minimizing the levels of lipid peroxidation and protein carbonyls in the early stages of development. The above indicates the importance for some marine invertebrates, which lack external physical protection such as shells or body plates, to seek shelter that minimizes UVB-induced photo-oxidative damage. In the present study, the sediment constituted a refuge and played an important role in the physical protection of both adults and juveniles of *A. hermaphroditica* against radiation, minimizing photo-oxidative damage.

A previous study in infaunal organisms (the amphipod *Chaetocorophium lucasi* and juvenile stages of the bivalves *Macomona liliana* and *Austrovenus stutchburyi*) showed that exposure to UVR in the absence of sediment generates between 5 and 10 times more phototoxicity (photoactivation of polycyclic aromatic hydrocarbons (PAH)) than in control organisms (without UVR). This indicates the high susceptibility of this type of organism to UVR radiation, and it is the sediment that acts as a physical barrier to radiation, which increases the survival of this type of animal by more than 50% [66].

In our study, the increased susceptibility to photo-oxidation in juveniles of *A. hermaphroditica* may be associated with acclimation to low levels of environmental radiation offered by the GVC during embryonic development, coupled with the presence of zooxanthellae that are acquired during the hatching process [67]. Internal incubation is part of the embryonic development strategy, which implies that the planulae larvae develop into tentaculate juveniles inside the GVC of an adult (Cubillos pers. obs.). This process is conceived as a strategy that allows embryonic care under high levels of environmental stress [68], which is consistent with the protection identified in this study when juveniles were confronted with UVR exposure.

A previous study carried out with the symbiont anemone *Anthopleura elegantissima* showed that the presence of zooxanthellae generates a negative phototactic response of the host anemone when exposed to high levels of radiation [69], a response that coincides with that of the symbiont anemone *A. hermaphroditica*. In photoautotrophic organisms, the photosynthetic apparatus represents the most delicate and fragile cellular structure that can be affected by UVB and UVA [70]. For example, chlorophyll levels in zooxanthellae of the symbiotic anemone *Aiptasia pallida* were significantly reduced due to photo-oxidation because of increased UVB levels [71], resulting in high levels of oxidative damage in the dinoflagellate *Symbiodinium bermudense*, reducing its photosynthetic capacity and impacting RUBISCO activity [72]. Previous studies indicated that overexposure of primary producers to UVB generates structural damage to the D1 protein of PSII, which is the molecule responsible for maintaining the photoprotection process in primary producers [73,74,75]. Although the spectral intensity of UVA is 10 times higher than that of UVB, it is also responsible for inducing photo-oxidative damage to PSII and PSI in photosynthetic organisms [76,77].

Therefore, when juvenile anemones of *A. hermaphroditica* leave the GVC where they are brooded, they must abruptly face a new environment with high levels of UVB and UVA, which could induce the high levels of photo-oxidative damage observed in this study. A similar situation was identified in the anemone *Actinia tenebrosa*, where the rapid transition from an environment with low levels of UVB to one with increased levels triggered the production of higher amounts of lipid peroxidation and protein carbonyl, increasing the basal levels by five and six times, respectively [50]. *A. hermaphroditica* adults, but mainly juveniles, increase the burial rate in the sediment, minimizing photo-oxidative damage when the substrate acts as a protective barrier for their tissues.

Levels of total antioxidant capacity observed in *A. hermaphroditica* during experimental exposure to UVB with sediment (68 mg Trolox Eq g^−1^ FW) are consistent with those observed (35 to 70 mg Trolox Eq g^−1^ FW) in the field for the same species in the Quempillén estuary [10]. Significant increases in total antioxidant capacity at PAB for adults and PA and PAB for juveniles reinforce the fact that juveniles of *A. hermaphroditica* show greater susceptibility to experimental radiation. Similar results were observed in the intertidal anemone *A. tenebrosa*, where controlled exposures to PA and PAB generated significant increases in GPox and GSH levels compared to those exposed only to P radiation [50]. Similarly, increases in SOD and CAT enzyme activity levels of 61% and 42% were recorded when pluteus larvae of *Sterechinus neumayerii* were exposed to P and PAB radiation for a period of 96 h [65]. Our results indicate that the level of antioxidant response in *A. hermaphroditica* was mainly associated with the developmental stage and the physical sediment protection with which they were provided. The sediment acted as a physical barrier for adults exposed to UVR by reducing total antioxidant defenses by 20–27% compared to those anemones exposed without sediment.

Although burial is a strategy to minimize UVB absorption damage, the use of photo-protective compounds such as mycosporine-like amino acids (MAAs) can minimize the effects of UVR exposure thanks to their high photo-stability since they can absorb wavelengths between 295 and 365 nm and dissipate the energy without the generation of ROS [78,79,80]. These secondary metabolites were widely used by sea anemones [81,82] as a photo-protective mechanism to minimize the generation of cyclobutane pyrimidine dimmers (CPDs) in the DNA [83], mutations that affect the replication and transcription of the genetic material [84]. The presence of MAAs in sea anemones was particularly associated with trophic transference and the presence of symbiotic zooxanthellae that are involved in the de novo synthesis of these compounds [82].

Although the photodynamic response of MAAs was not analyzed in this study, it is necessary to indicate that in *A. hermaphroditica*, the presence of Mycosporine 2-glyicine (Myc 2-gly) was described, which can absorb mostly λ_max_ = 334 nm, a compound that would allow minimizing the damage due to UVR absorption [10]. Previous observations carried out in sea anemones of the *Anthopleura* genus indicate that the presence of Myc 2-gly is common among different species, and their presence is associated with those that exhibit symbiotic relations with zooxanthellae [82]. Considering that *A. hermaphroditica* is a symbiont anemone [30], the transfer of zooxanthellae to hatched juveniles inside the GVC (Cubillos pers. obs.) could provide some degree of protection when facing a new environment characterized by high levels of UVR radiation. However, a previous study indicates that the density of zooxanthellae in the larvae of the sea anemone *A. elegantissima* is directly related to the radiation levels to which they were experimentally exposed [85]. Thus, low levels of radiation within the GVC of adults of *A. hermaphroditica* semi-buried in the sediment would generate reduced levels of zooxanthellae infestation in newly hatched juveniles. Consequently, this may induce a low level of MAAs in juvenile tissues, increasing the levels of molecular damage when exposed to increased levels of UVR radiation.

Consequently, higher levels of oxidative damage in newly hatched juveniles compared to adults of *A. hermaphroditica* exposed to UVB would have serious implications for the energy budget since the de novo synthesis of enzymes and antioxidant compounds implies a high demand for ATP [86,87]. The above involves an imbalance in the energy budget, affecting the cellular processes destined to cell repair activities, as well as the degradation and synthesis of new cell components damaged by the photo-oxidative effect [88,89,90]. Additional studies are needed to determine whether the body walls of adult *A. hermaphroditica* can protect incubated embryos against UVB when they are removed from the sediment by bioturbation processes.

## 5. Conclusions

The present study showed that the burrowing activity of the intertidal anemone *A. hermaphroditica* is a behavioral response to UVR radiation. Independent of the developmental stage, PAR + UVA + UVB radiation caused adults and juveniles to conceal their bodies by burrowing faster into the sediment compared to those exposed only to PAR radiation, providing evidence that these anemones are evading UVB, reducing oxidative damage. A general pattern of adult and juvenile anemones exposed to PAB is to suffer higher levels of oxidative damage than those generated by P radiation treatment alone, indicating its noxious effect. Although the presence or absence of sediment generated a clear photo-oxidative response in adults under each radiation treatment, the total antioxidant capacity was significantly higher when anemones were exposed to different radiations without sediment. The presence of sediment seems to be important to juvenile anemones, especially when they are exposed to UVB radiation, possibly due to their elevated photosensitivity, considering that naturally, they develop inside of the GVC of burrowed adults, a microenvironment characterized by reduced or null radiation conditions. Further studies should focus on understanding the effect of brooding in the GVC and its role in protecting embryos against UVB radiation.

## Figures and Tables

**Figure 1 antioxidants-11-01725-f001:**
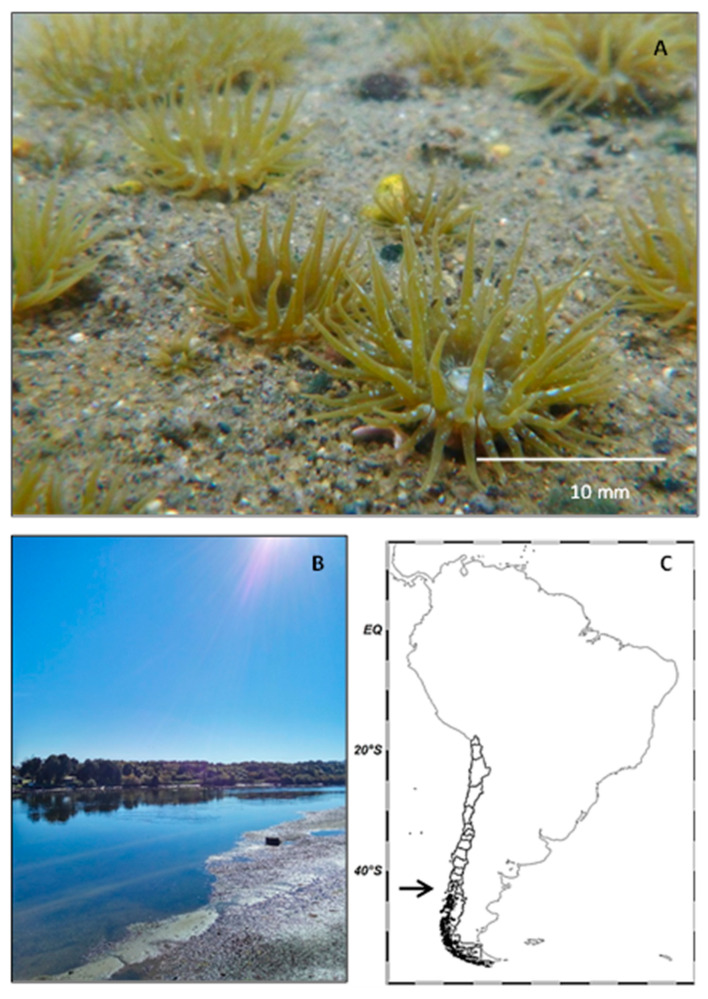
Adult of the estuarine anemone *Anthopleura hermaphroditica* (**A**) collected from Quempillén estuary (41°520 S; 73°460 W) of Chiloé Island (**B**), Southern Chile (**C**). Arrow shows location of Chiloé Island in Chile. Map generated using Ocean Data View software (version 5.6.2, Schlitzer, Reiner, Ocean Data View, odv.awi.de, 2021, Germany).

**Figure 2 antioxidants-11-01725-f002:**
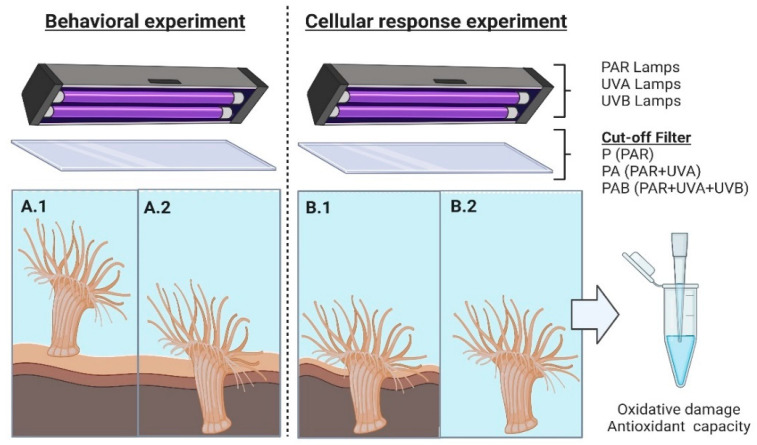
Experimental setup for behavioral ((**A.1**) initial time/(**A.2**) final time) and cellular responses ((**B.1**) with sediment/(**B.2**) without sediment) in adults and juveniles of the sea anemone *A. hermaphroditica* exposed to different radiation conditions generated by a combination of fluorescent lamps (PAR, UVA and UVB) and cut-off filters (P: PAR, PA: PAR + UVA and PAB: PAR + UVA + UVB). Figure created using BioRender.com (accessed on 9 August 2022).

**Figure 3 antioxidants-11-01725-f003:**
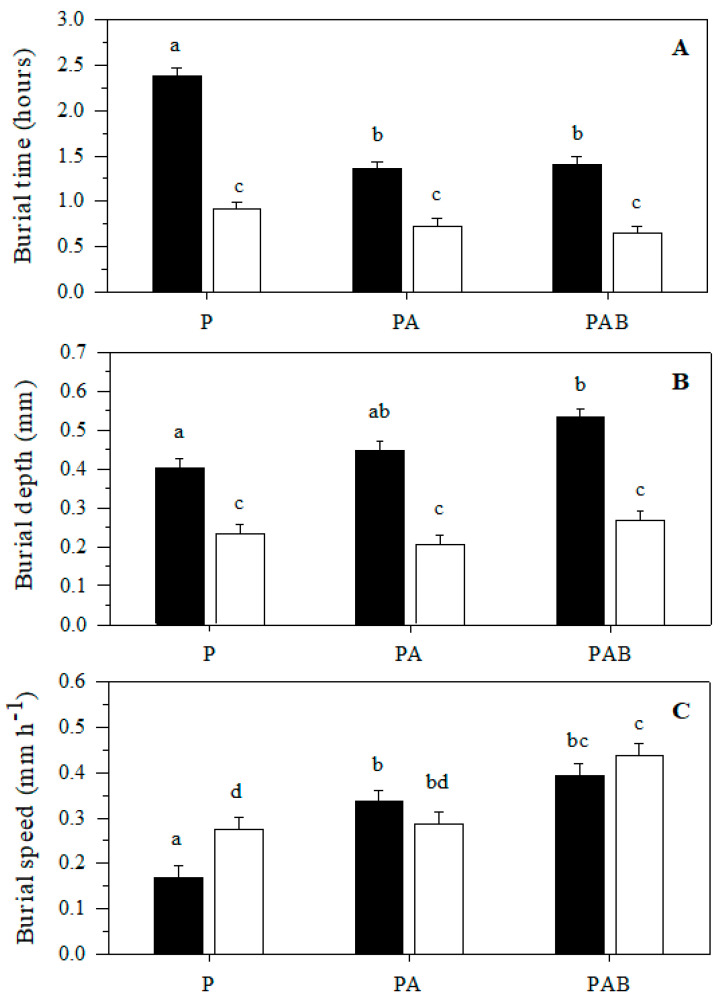
Burial time (**A**), burial depth (**B**) and burial speed (**C**) of adults (black bars) and juveniles (white bars) of the sea anemone *A. hermaphroditica* in the sediment during exposition to P (PAR), PA (PAR + UVA) and PAB (PAR + UVA + UVB) radiation treatment. Bars show mean ± SE. Different letters over bars indicate significant differences (*p* < 0.05).

**Figure 4 antioxidants-11-01725-f004:**
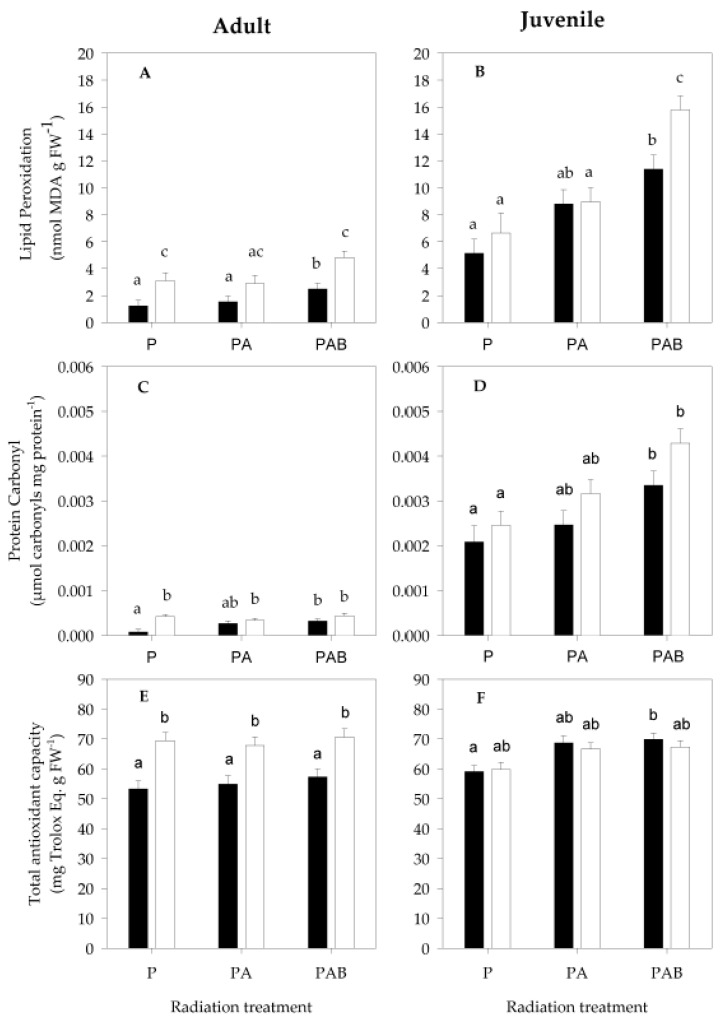
Protein carbonyl (**A**,**B**), lipid peroxidation (**C**,**D**) and total antioxidant capacity (**E**,**F**) in adults and juveniles of the anemone *A. hermaphroditica* exposed to P (PAR), PA (PAR + UVA) and PAB (PAR + UVA + UVB) radiation treatments with sediment (black bars) and without sediment (white bars). Error bars indicate ±SE. Different letters over bars indicate significant differences (*p* < 0.05).

**Table 1 antioxidants-11-01725-t001:** Two-way ANOVA for comparison of burial parameters (time, depth and speed) in different developmental stages (adults and juveniles) of *A. hermaphroditica* exposed to different levels of experimental radiation P (PAR), PA (PAR + UVA) and PAB (PAR + UVA + UVB). Bold numbers indicate significant difference (*p* < 0.05).

Source of Variation	df	ms	f	*p*
Burial time				
Developmental stage (DS)	1	15.130	212.389	**<0.001**
Radiation treatment (RT)	2	2.783	39.066	**<0.001**
DS × RT	2	1.119	15.71	**<0.001**
Error	60	0.0712		
Burial depth				
Developmental stage (DS)	1	1.370	171.89	**<0.001**
Radiation treatment (RT)	2	0.0663	8.311	**0.001**
DS × RT	2	0.0152	1.913	0.157
Error	60	0.00797		
Burial speed				
Developmental stage (DS)	1	0.0190	2.514	0.118
Radiation treatment (RT)	2	0.206	27.244	**<0.001**
DS × RT	2	0.0338	4.481	**0.015**
Error	60	0.00755		

**Table 2 antioxidants-11-01725-t002:** Three-way ANOVA for the estimation of oxidative response (protein carbonyl and lipid peroxidation) and antioxidant capacity in adults and juveniles of the anemone *A. hermaphroditica* exposed to different levels of experimental radiation P (PAR), PA (PAR + UVA) and PAB (PAR + UVA + UVB) with and without sediment. Bold numbers indicate significant difference (*p* < 0.05).

Source of Variation	df	ms	f	*p*
Lipid peroxidation				
Developmental stage (DS)	1	431.96	158.216	**<0.001**
Sediment condition (SC)	1	35.15	12.875	**0.001**
Radiation treatment (RT)	2	70.90	25.971	**<0.001**
DS × SC	1	0.070	0.0258	0.873
DS × RT	2	30.36	11.121	**<0.001**
SC × RT	2	6.093	2.232	0.125
DS × SC × RT	2	2.257	0.925	0.408
Residual	29	2.730		
Protein carbonyl				
Developmental stage (DS)	1	0.0000765	351.271	**<0.001**
Sediment condition (SC)	1	0.00000193	8.848	**0.006**
Radiation treatment (RT)	2	0.00000260	11.934	**<0.001**
DS × SC	1	0.00000062	3.043	0.091
DS × RT	2	0.00000186	8.564	**<0.001**
SC × RT	2	0.0000000305	0.140	0.870
DS × SC × RT	2	0.000000152	0.700	0.504
Residual	32	0.000000218		
Total antioxidant capacity				
Developmental stage (DS)	1	102.022	4.221	**<0.048**
Sediment condition (SC)	1	463.94	19.191	**<0.001**
Radiation treatment (RT)	2	137.22	5.677	**0.007**
DS × SC	1	674.15	27.891	**<0.001**
DS × RT	2	69.96	2.895	0.069
SC × RT	2	11.307	0.468	0.630
DS × SC × RT	2	0209	0.00864	0.991
Residual	34	25.634		

## Data Availability

The data presented in this study are available in a data repository.

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
