# Peer review of "Effects of Ultraviolet Radiation on Sediment Burial Parameters and Photo-Oxidative Response of the Intertidal Anemone Anthopleura hermaphroditica"

_antioxidants, 2022, doi:10.3390/antiox11091725_

Round 1
Reviewer 1 Report
This is an interesting paper concerning with the effects of Ultraviolet Radiation on Sediment Burial Parameters 2 and Photo-Oxidative Response of the Intertidal Anemone Anthopleura hermaphroditica from the pacific coast of southern Chile. The manuscript is clearly written and my principal concerns are focused on two issues:
- On the one hand, anemones of the genus Anthopleura have attached gravel to their columns that can protect them from external factors such as UV radiation. In the image provided by the authors the anemones have not the attached gravel, but I do not if this the natural condition of this species or perhaps the cover is missing due to stress). This should be further discussed in the ms. For example, it is mentioned that juveniles bury themselves more quickly than adults, a possible explanation could be their lower gravel coverage in their columns.
On the other hand, In my opinion the importance of the mycosporine-like amino acids (MAAs) is underestimated. I suggest discussing in more detail the possible role of MAAs in protecting against UV radiation, and subsequent molecular damage. There is relevant literature on the presence of MAAs in intertidal anemones in the northern hemisphere (Shick et al. 2002. Biol. Bull. 203, 315–330) and southern hemisphere (Arbeloa et al. 2010. Comp. Bioch. Physiol. B, 156: 216-221.) that should be included and discussed in the ms. Again the more susceptibility of juveniles to molecular damage by UV radiation can hypothetically due to a minor concentration of MAAs in their tissues.
Reviewer 2 Report
The Paper “Effects of Ultraviolet Radiation on Sediment Burial Parameters and Photo-Oxidative Response of the Intertidal Anemone Anthopleura hermaphroditica” reports important data about this model organism and the specific response to radiation. The analysis was carried out studying quantitative data on oxidative stress. A particular and interesting focus was about observations with and without sediments.
I suggest minor revision:
1. Introduction
Authors should provide more data about the importance and role of this model system and the validity of the studies conducted on radiation (environmental risks and dangers).
2. Materials and Methods
More technical information on radiation sources should be reported.
3. Results.
In figure 3 I suggest to report P, PA and PAB also in histograms A and B.
Apart from the above suggestions, I do not find any objection to giving my favorable opinion for the publication of this work on Antioxidants journal.
